# SARS-CoV-2 spike and nucleocapsid proteins fail to activate human dendritic cells or γδ T cells

**Kiran Singh[1], Sita Cogan[1], Stefan Elekes[1], Dearbhla M. Murphy[1], Sinead Cummins[1], Rory Curran[1], Zaneta Najda[2], Margaret R. Dunne[1], Gráinne Jameson[1], Siobhan Gargan[3], Seamus Martin[2], Aideen Long[3], Derek G. Doherty[1]***

**1** Discipline of Immunology, Trinity Translational Medicine Institute, Trinity College Dublin, St. James's Hospital, Dublin, Ireland, **2** Molecular Cell Biology Laboratory, Smurfit Institute of Genetics, Trinity College Dublin, Dublin, Ireland, **3** Discipline of Clinical Medicine, Trinity Translational Medicine Institute, Trinity College Dublin, St. James's Hospital, Dublin, Ireland

\* derek.doherty@tcd.ie

## Abstract

γδ T cells are thought to contribute to immunity against severe acute respiratory syndrome coronavirus 2 (SARS-CoV-2), but the mechanisms by which they are activated by the virus are unknown. Using flow cytometry, we investigated if the two most abundant viral structural proteins, spike and nucleocapsid, can activate human γδ T cell subsets, directly or in the presence of dendritic cells (DC). Both proteins failed to induce interferon-γ production by Vδ1 or Vδ2 T cells within fresh mononuclear cells or lines of expanded γδ T cells generated from healthy donors, but the same proteins stimulated CD3+ cells from COVID-19 patients. The nucleocapsid protein stimulated interleukin-12 production by DC and downstream inter-feron-γ production by co-cultured Vδ1 and Vδ2 T cells, but protease digestion and use of an alternative nucleocapsid preparation indicated that this activity was due to contaminating non-protein material. Thus, SARS-CoV-2 spike and nucleocapsid proteins do not have stimulatory activity for DC or γδ T cells. We propose that γδ T cell activation in COVID-19 patients is mediated by immune recognition of viral RNA or other structural proteins by γδ T cells, or by other immune cells, such as DC, that produce γδ T cell-stimulatory ligands or cytokines.

**Data Availability Statement:** All relevant data are within the manuscript and its Supporting information files.

## Introduction

Severe acute respiratory syndrome coronavirus 2 (SARS-CoV-2), the pathogen responsible for the coronavirus disease 2019 (COVID-19) pandemic, has infected over 500 million people in the 2 years since its first appearance in December 2019, leading to over 6 million deaths world-wide [1]. Infected individuals experience a wide spectrum of clinical manifestations ranging from asymptomatic infection, through mild respiratory symptoms, to severe respiratory insuf-ficiency and extra-pulmonary manifestations which can lead to organ failure and death [2, 3]. The host immune response to SARS-CoV-2 plays a significant role in determining the

**Funding:** SG and ZN were funded by SCIENCE FOUNDATION IRELAND, grant number SFI-20/SPP/3685. The funders had no role in study design, data collection and analysis, decision to publish, or preparation of the manuscript.

**Competing interests:** The authors have declared that no competing interests exist.

outcome of COVID-19. Severe disease is associated with macrophage and neutrophil activation and recruitment to the lungs, and elevated serum levels of chemokines and cytokines, including interleukin-1β (IL-1β), IL-6, tumor necrosis factor-α (TNF-α) and interferon-γ (IFN-γ)-induced protein 10 (IP-10; also known as CXCL10) [4–7]. Severe COVID-19 is also associated with depletions and functional exhaustion of circulating CD4 T cell, CD8 T cells, natural killer (NK) cells, B cells, follicular helper T cells and innate T cells such as γδ T cells, mucosa-associated invariant T (MAIT) cells and natural killer T (NKT) cells [6–8].

Human γδ T cells contribute to immunity against viruses as evidenced by their potent cytotoxic activities and rapid secretion of cytokines, such as IFN-γ and TNF-α, which stimulate anti-viral adaptive immune responses [9, 10]. They can kill cells infected with cytomegalovirus [11] and influenza virus [12] and can suppress replication of human immunodeficiency virus, hepatitis B and C viruses, herpesviruses, picornaviruses and others [13–15]. Evidence for a protective role for the Vδ2 subset of γδ T cells against SARS-CoV, the causative agent of the 2003 coronavirus epidemic, is provided by Poccia and co-workers [16]. In this study, convalescent patients exhibited expansions of Vδ2 T cells, which killed SARS-CoV-infected cells, and inhibited SARS-CoV replication via non-cytolytic mechanisms.

Immunophenotyping studies on patients with COVID-19 have demonstrated that γδ T cells are depleted from the blood of patients with severe disease [6, 7, 17, 18]. Human γδ T cells comprise 2 major structural subsets based on their T cell receptor (TCR) δ-chain usage and a number of minor subsets. Vδ2 T cells are predominant in the blood, while Vδ1 T cells predominate in tissues, such as the gut, liver and dermis [9, 10]. The Vδ2 subset, only, is depleted from the blood of patients with severe COVID-19, while Vδ1 T cell numbers are unchanged [6, 7]. Circulating and pulmonary γδ T cells from patients with severe COVID-19 display effector-memory phenotypes, expressing CD25, CD69 and HLA-DR [7, 19, 20]. They also express markers of exhaustion (PD-1) and senescence (CD57) and exhibit impaired IFN-γ production upon pharmacological stimulation *in vitro* [7, 21]. The depletions of circulating γδ T cells from the blood are likely due to their migration to the lungs, as these cells have been shown to be abundant in the lungs of patients with severe COVID-19 [7, 22]. Because of their potential involvement in immunity against SARS-CoV-2, and possibly in the inflammatory events that drive severe COVID-19, γδ T cells may serve as targets for immunotherapy. Cellular immunotherapies using γδ T cells are currently being tested in the setting of cancer [23, 24] and have been proposed for the treatment of infectious diseases, including human immunodeficiency virus [25, 26], respiratory syncytial virus [27] and SARS-CoV-2 [28].

The mechanisms by which γδ T cells are activated in response to SARS-CoV-2 infection are poorly understood. γδ T cells are equipped with multiple stimulatory receptors, including the TCR, toll-like receptors (TLR) [29, 30], natural killer group 2D (NKG2D), DNAX accessory molecule-1 (DNAM-1), and natural cytotoxicity receptors (NCR) [10, 13, 31], which could potentially recognize viral components directly, or ligands produced by cells in response to viral infection. γδ T cells also express Fc receptors [10] which may bind opsonised virus, leading to activation. γδ T cells can also be activated by cytokines, such as IL-12, IL-15, IL-23 or TNF-α, released by myeloid cells such as dendritic cells (DC) [32, 33].

DCs can sense SARS-CoV-2 genomic RNA via pathogen receptors, including retinoic acid-inducible gene I (RIG-I), melanoma differentiation-associated protein 5 (MDA5) [34] and TLRs 3, 7 and 8 [35–37]. Furthermore, host cell DNA in the cytoplasm, which is generated in response to SARS-CoV-2-infection can be detected by cyclic GMP–AMP synthase (cGAS) [38]. We hypothesized that viral structural proteins may also trigger the activation of γδ T cells, either directly or via the activation of cells of the innate immune system. The spike protein of SARS-CoV-2 has been reported to bind to TLR4 [39, 40] on macrophages, leading to their activation, whereas the envelope protein can mediate inflammatory responses by binding

to TLR2 [41]. We investigated if either of the two most abundant SARS-CoV-2 structural proteins, spike and nucleocapsid, which provide immunogenic peptides for recognition by conventional T cells [42, 43], can activate Vδ1 and Vδ2 T cells, either directly or in the presence of DC. The results show that these SARS-CoV-2 proteins do not intrinsically induce maturation or cytokine production by DC or activate either subset of γδ T cells.

## Materials and methods

### Blood samples

EDTA-anticoagulated blood samples were obtained from healthy volunteers, from used anonymous buffy coat packs obtained from the Irish Blood Transfusion Service, and from two patients with COVID-19 attending St. James's Hospital, Dublin. Ethical approval for this project was obtained from the Research Ethics Committees of St. James's Hospital and Trinity College Dublin and the study was carried out following the rules of the Declaration of Helsinki of 1975, revised in 2013. Informed written consent was obtained from the two patients with COVID-19. Consent for the use of anonymous buffy coat packs was not required. Peripheral blood mononuclear cells (PBMC) were prepared from all blood samples by standard density gradient centrifugation over Lymphoprep (StemCell Technologies).

### Generation of DC

Monocytes were isolated from $2x10^8$ PBMC by positive selection using CD14 Microbeads (Miltenyi Biotec) and resuspended in 'DC medium' (RPMI 1640 with Glutamax containing 25 mM HEPES, 50 μg/ml streptomycin, 50 U/ml penicillin and 10% heat-inactivated, Hyclone low-endotoxin fetal bovine serum (FBS)) at a final density of $1x10^6$ cells/mL. Monocytes were cultured for 6 days in 6-well plates at 37˚C, 5% $CO_2$ in the presence of 50 ng/mL granulocyte macrophage colony stimulating factor (GM-CSF) and 70 ng/mL IL-4 to allow differentiation into immature DC, replacing the medium with fresh medium containing the cytokines after 3 days [44]. Flow cytometry was used to verify that differentiation into immature DC had taken place and cells expressed HLA-DR and CD11c and no longer expressed CD14.

### Generation of primary cultures of γδ T cells

Primary cultures of γδ T cells, which contained Vδ1 and Vδ2 T cells, were generated from buffy coat packs, obtained from 500 mL blood, which yielded up to $10^9$ PBMC. The PBMC were suspended in RPMI medium with Glutamax supplemented with 10% FBS and seeded at a density of $4x10^6$ cells/mL in T75 flasks and maintained at 37˚C and 5% $CO_2$ overnight. α/βTCR$^+$ T cells were then depleted from $2x10^8$ PBMC by magnetic depletion using a CliniMACS TCR α/β-Biotin kit, as per the supplier's instructions (Miltenyi Biotec). The α/β T cell-depleted fraction was then resuspended at a density of $7.5x10^6$ cells/mL in 'γδ T cell medium' (RPMI Glutamax medium containing 10% FBS, 25 mM HEPES, 1% PenStrep, 1 mM sodium pyruvate, 50 μM 2-mercaptoethanol, 1% non-essential amino acids and 1% essential amino acids) and stimulated with 1 μg/mL anti-CD3 antibody (clone OKT3, BioLegend). Cells were cultured in 96-well round-bottom plates at 37˚C and 5% $CO_2$. After 24 hours, and every 3–4 days thereafter, the medium was replaced with fresh medium containing 100 U/mL IL-2 and 70 ng/ml of IL-15 (Miltenyi Biotec). Cells were cultured for up to 60 days and the purities and relative frequencies and phenotypes of Vδ1 and Vδ2 T cells were assessed by flow cytometry. At the time of use in experiments, Vδ1 T cells accounted for 2.1–37% (mean 22.5%) of the cells, whereas Vδ2 T cells accounted for 53.2–97.2% (mean 72.4%) and Vδ3 T cells accounted for 0.3–4.3% (mean 2.1%). The remaining cells were Vδ1$^-$ Vδ2$^-$ Vδ3$^-$ γδ T cells and no αβ T cells, B cells or

NK cells were present. The mean CD4/CD8 ratios of the Vδ1, Vδ2 and Vδ3 T cells were 3.2, 4.2 and 3.8. respectively and 38.8%, 71.0% and 91.1% of these cells expressed NKG2D.

## Recombinant SARS-CoV-2 proteins and peptides

Pools of overlapping peptides corresponding to the immunodominant epitopes (Peptivators) of the spike protein (S peptide), the S1 domain of the spike protein (S1 peptide), the S2 domain (S+ peptide), or the nucleocapsid protein (N peptide) were purchased from Miltenyi Biotec. Recombinant SARS-CoV-2 spike trimer with a C-terminal His-tag, produced in HEK293 cells, was purchased from Peak Proteins Ltd. The protein was purified by nickel affinity chromatography, followed by size exclusion chromatography. Recombinant SARS-CoV-2 nucleocapsid, also expressed in HEK293 cells and purified by metal ion affinity chromatography, was purchased from RayBiotech and denoted 'RB-nucleocapsid'. A second recombinant SARS-CoV-2 nucleocapsid preparation, denoted 'SM-nucleocapsid' was expressed in an endotoxin-free strain of *Escherichia coli* (ClearColi BL21 [DE3]). Bacteria were transformed with a pET28a(+) plasmid encoding SARS CoV-2 nucleocapsid protein fused to an in frame C-terminal AAALE linker and a 6xHis tag (kindly provided by the Joshua-Tor laboratory, Cold Spring Harbor, New York, USA). Bacterial transformants were grown in Kan[+] LB medium and nucelocapsid protein was induced through addition of 100 mM isopropyl ß-D-1-thiogalactopyranoside for 3 h at 37˚C. Bacteria were lysed in 50 mM Tris-HCl (pH 8.5), 10 mM 2-mercaptoethanol, 1 mM phenylmethylsulfonyl fluoride (PMSF), 2 µg/ml aprotonin, 10 µg/ml leupeptin and, after clarification of bacterial lysates by centrifugation of insoluble material, nucleocapsid protein was captured on Ni-NTA agarose beads and eluted into phosphate buffered saline, pH 7.2, containing 100 mM imidazole, 1 mM phenylmethylsulfonyl fluoride, 2 µg/ml aprotonin and 10 µg/ml leupeptin.

## Direct stimulation of PBMC with SARS-CoV-2 spike and nucleocapsid proteins and peptides

Total PBMC or expanded lines of γδ T cells (0.25x10[6] cells) were stimulated in 96-well round bottom plates for 6 hours or overnight with 1 µg/mL whole spike protein or 1 µg/mL of overlapping peptides corresponding to the immunodominant epitopes of the spike or nucleocapsid proteins. Stimulation with 50 ng/mL phorbol myristate acetate with 1 µg/mL ionomycin (PMA/I) served as a positive control. Brefeldin A (10 µg/mL) was added for the last 4 hours of stimulation to prevent cytokine secretion. Cells were then analysed for intracellular expression of cytokines as described below.

## *In vitro* stimulation of DC in the absence and presence of γδ T cells

0.25x10[6] DC were seeded in duplicate in 96-well round bottom plates and stimulated with medium alone, lipopolysaccharide (LPS; 100 ng/mL), or 1 µg/mL spike or nucleocapsid protein. In some experiments, the spike and nucleocapsid proteins were first digested with 1 µg/mL V8 protease from *Staphylococcus aureus* (MP Biochemicals). After 4 hours, 0.25x10[6] expanded γδ T cells were added to one set of stimulated DC. After a further 4 hours, 10 µg/mL brefeldin A was added and the cells were cultured overnight. Cells were then analysed for intracellular expression of cytokines as described below.

## Analysis of cell activation

PBMC, expanded γδ T cells, DC or DC-γδ T cell co-cultures were stained with a dead cell stain (efluor 506-conjugated Fixable Viability Dye; Thermofisher) and Fc receptors were then

blocked using Fc Blocking Reagent (Miltenyi Biotec). Cells were then stained with fluoro-chrome-conjugated monoclonal antibodies (mAbs) specific for CD3, Vδ1, Vδ2, CD11c, HLA-DR, CD83, CD86. Cells were then fixed and permeabilised using the Fixation and Permeabilization Solutions (BD Biosciences) and stained using fluorochrome-conjugated mAbs specific for IL-12 (p40/p70), TNF-α and IFN-γ. MAbs were purchased from Miltenyi Biotec, BD Biosciences and BioLegend. The cells were then acquired on a FACSCanto II flow cytometer (Becton Dickinson) and data were analysed using FlowJo software (Tree Star). Fig 1 shows the gating strategy used to measure the mean fluorescence intensities (MFI) of HLA-DR, CD40, CD83 and CD86 expression by $CD11c^+$ cells (DC) and the percentages of $CD11c^+$ cells that produced IL-12 and TNF-α (Fig 1A) and the percentages of $CD3^+$ cells (T cells), Vδ1 T cells and Vδ2 T cells that produced IFN-γ and TNF-α (Fig 1B).

## Statistical analysis

GraphPad Prism 9.1.1 was used to analyse and plot results generated from Flow Cytometry Software (FCS) file data. Results in treatment groups were compared using the Wilcoxon signed rank test or the paired t-test where data were normally distributed. *P*-values of <0.05 are denoted *, whereas ** denotes P<0.01; ***denotes P<0.005 and **** denotes P<0.001.

# Results

## SARS-CoV-2 spike and nucleocapsid proteins do not directly activate γδ T cells to produce IFN-γ

PBMC from 3 healthy donors and expanded γδ T cells from 4 donors were treated for 6 hours with medium alone, PMA/I, or recombinant SARS-CoV-2 spike or nucleocapsid proteins and the production of IFN-γ by gated Vδ1 and Vδ2 T cells was investigated by flow cytometry. While PMA/I stimulation induced significant IFN-γ production by both Vδ1 (Fig 2A) and Vδ2 (Fig 2B) T cells within PBMC and within expanded γδ T cells, neither spike nor nucleocapsid protein induced IFN-γ production above background levels. The same spike protein induced IFN-γ production by 0.06 and 0.08% of $CD3^+$ T cells from 2 patients with COVID-19 (Fig 2C), indicating that this protein is capable of stimulating conventional T cells. IFN-γ production by $CD3^+$ T cells within PBMC stimulated with the nucleocapsid protein was not tested.

## SARS-CoV-2 spike and nucleocapsid proteins do not induce DC maturation *in vitro*

DC from 4 donors were stimulated overnight with medium alone, LPS, or recombinant SARS-CoV-2 spike or RB-nucleocapsid proteins and the induction of the maturation markers HLA-DR, CD40, CD83 and CD83 by $CD11c^+$ cells (DC) was measured by flow cytometry. The fold increases in mean fluorescence intensities (MFI) of these markers are shown in Fig 3A. LPS treatment led to weak induction of all markers on DC, but this was only significant for HLA-DR. Treatment with the spike protein had no effect HLA-DR, CD40, CD83 or CD86 expression by DC. Interestingly, the RB-nucleocapsid protein induced mean 2-4-fold increases in the MFI of all HLA-DR and CD83, however, these increases were not significant using the Wilcoxon Signed Rank test.

## SARS-CoV-2 RB-nucleocapsid but not spike protein induces IL-12, but not TNF-α production by DC

We also measured the percentages of $CD11c^+$ cells that produced IL-12 and TNF-α in response to treatment with medium, LPS, spike or RB-nucleocapsid proteins. Since γδ T cells

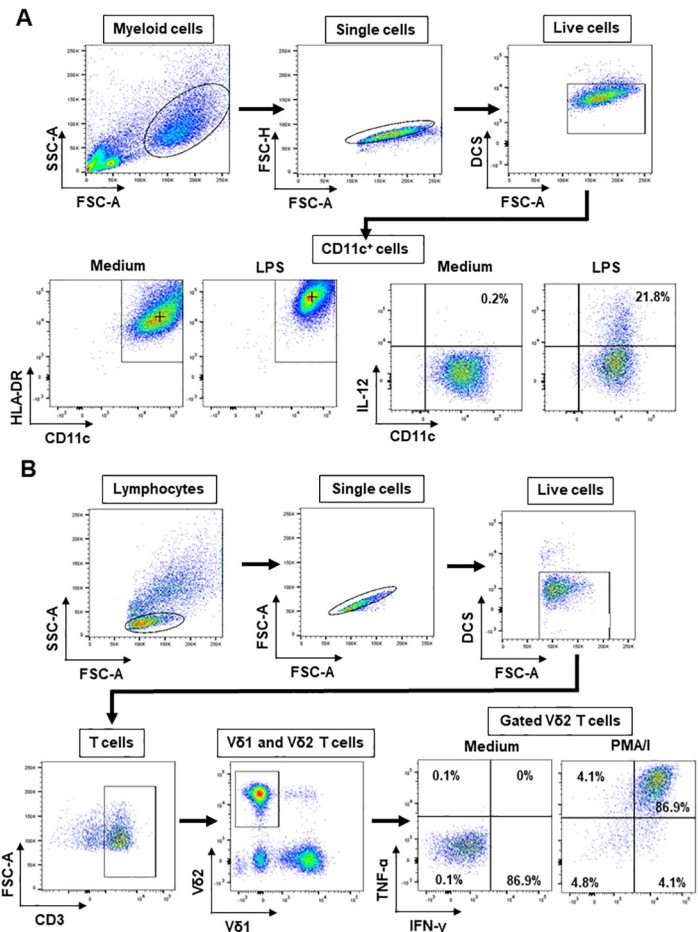

**Fig 1. Gating strategy for the detection of activated and cytokine-producing dendritic cells (DC), Vδ1 and Vδ2 T cells by flow cytometry. A**, Monocyte-derived DC or DC-γδ T cell co-cultures were stained with a dead cell stain (DCS) and monoclonal antibodies (mAb) specific for cell-surface CD3, CD11c, HLA-DR, CD40, CD83 and CD86 and intracellular IFN-γ, IL-12 and TNF-α and analysed by flow cytometry. Upper panels, left to right: flow cytometry dot plots showing forward scatter area (FSC-A) plotted against side scatter area (SSC-A) with an electronic gate drawn around the DCs; Dot plot showing FSC-A plotted against FSC-height (FSC-H) for gated DCs with a gate drawn around the single cells; Dot plot showing FSC-A plotted against the dead cell stain for gated single cells with a gate drawn around the live cells. Lower panels: Dot plots showing expression of HLA-DR (left) and IL-12 (right) by CD11c + cells after stimulation with medium alone or LPS, used to determine the mean fluorescence intensity (MFI) of HLA-DR expression by DC (marked +) and the percentages of DC that produced IL-12. MFI of CD40, CD83 and CD86 and % positivity for TNF-α was similarly determined. **B**, γδ T cells or DC-γδ T cell co-cultures were stained with a dead cell stain (DCS) and mAbs specific for cell-surface CD3, Vδ1, Vδ2 and intracellular IFN-γ and TNF-α and analysed by flow cytometry. Upper panels, left to right: flow cytometry dot plots showing FSC-A plotted against SSC-A with an electronic gate drawn around the lymphocytes; Dot plot showing FSC-A plotted against FSC-H for gated lymphocytes with a gate drawn around the single cells; Dot plot showing FSC-A plotted against the dead cell stain for gated single cells with a gate drawn around the live cells. Lower panels, left to right: Dot plot showing expression of CD3 and FSC-A by single live lymphocytes with a gate drawn around the T cells; Dot plot showing Vδ1 and Vδ2 expression by gated CD3+ cells; Dot plot showing IFN-γ and TNF-α expression by gated Vδ2 T cells treated with medium or PMA and ionomycin (PMA/I) used to determine the percentage of Vδ2 T cells that produced each cytokine. Cytokine production by Vδ1 T cells was similarly determined.

can potently stimulate maturation and IL-12 production by DC [44–50], these experiments were carried out both in the absence and presence of equal numbers of expanded γδ T cells. Fig 3B shows that unstimulated DC did not produce IL-12, even when γδ T cells were present. Upon stimulation with LPS, a mean of 12% of DC, cultured alone, produced IL-12 and this

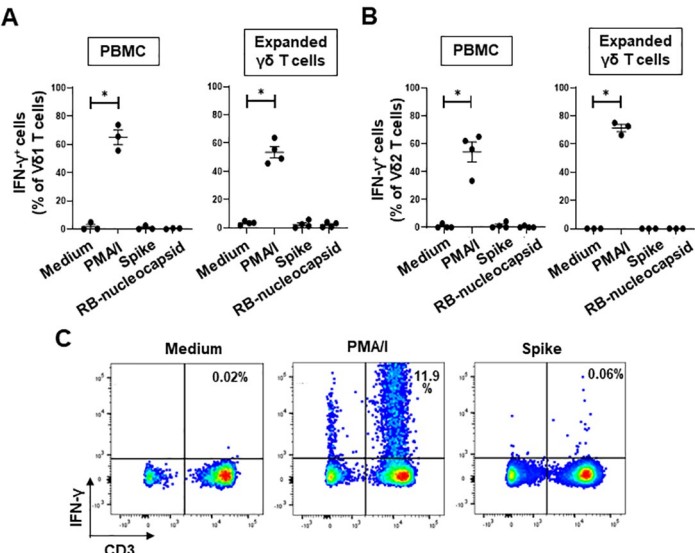

**Fig 2. SARS-CoV-2 spike and RB-nucleocapsid proteins do not directly activate Vδ1 or Vδ2 T cells within PBMC or expanded γδ T cells to produce IFN-γ in vitro. A and B**, PBMC from 3 donors and expanded γδ T cells from 4 donors were stimulated with medium alone, PMA and ionomycin (PMA/I) and SARS-CoV-2 spike and RB-nucleocapsid proteins and the percentages of Vδ1 (A) and Vδ2 (B) T cells that produced IFN-γ were measured by flow cytometry. Results are compared using the Wilcoxon signed rank test; *P<0.05. **C**, Representative flow cytometry dot plots showing PBMC from a COVID-19 patient after stimulation with medium, PMA/I and SARS-CoV-2 spike protein showing the frequencies of CD3+ T cells that produced IFN-γ.

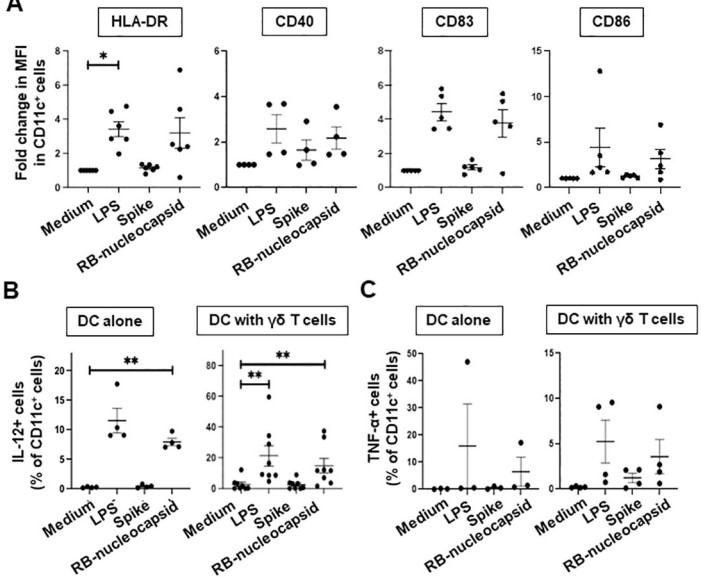

**Fig 3. Effect of SARS-CoV-2 spike and RB-nucleocapsid proteins on maturation of dendritic cells (DC), and production IL-12 and TNF-α by DC cultured alone and in the presence of γδ T cells. A**, Monocyte-derived DC from 4–6 donors were stimulated overnight with medium, LPS, spike and RB-nucleocapsid protein and the mean fluorescence intensity (MFI) of HLA-DR, CD40, CD83 and CD86 expression by CD11c+ DC was measured by flow cytometry. Scatter plots show the fold changes in MFI over unstimulated DC. **B and C**, DC cultured alone (n = 3–4) or in the presence of γδ T cells (n = 8), were stimulated as above and the percentages of DC that produced IL-12 (B) and TNF-α (C) were determined by flow cytometry. Frequencies were compared using the Wilcoxon signed rank test or the paired t-test where data were normally distributed. *P<0.05 and **P<0.01 compared to medium only.

frequency increased to 22% of DC when γδ T cells were present (P<0.01). Treatment with SARS-CoV-2 spike protein did not induce IL-12 production in the absence or presence of γδ T cells. However, treatment with RB-nucleocapsid protein led to significant induction of IL-12 production by DC, both cultured alone and in the presence of γδ T cells (*P*<0.01). In contrast, TNF-α production by DC was not affected by treatment with either the spike or nucleocapsid proteins ([Fig 3C]). These results suggest that SARS-CoV-2 nucleocapsid protein may have some stimulatory activity for DC, which is potentiated by γδ T cells.

## DC treated with SARS-CoV-2 RB-nucleocapsid but not spike protein induce IFN-γ production by Vδ1 and Vδ2 T cells

γδ T cells from 4–8 donors were co-cultured overnight with allogeneic monocyte-derived DC that had been treated for 4 hours with medium or SARS-CoV-2 spike or RB-nucleocapsid protein. PMA/I-treated γδ T cells served as positive controls. The percentages of Vδ1 and Vδ2 T cells that produced IFN-γ and TNF-α in response to each treatment were measured by flow cytometry. [Fig 4] shows that DC treated with RB-nucleocapsid, but not spike, protein induced IFN-γ production by a significant proportion of Vδ1 and Vδ2 T cells (*P*<0.05). In contrast, neither of the proteins induced TNF-α production by either γδ T cell subset.

## Activation of DC and γδ T cells by RB-nucleocapsid is mediated by a non-protein contaminant

The above experiments provide evidence that RB-nucleocapsid can induce IL-12 production by DC and that RB-nucleocapsid treated DC can induce IFN-γ production by Vδ1 and Vδ2 T cells, suggesting that SARS-CoV-2 nucleocapsid protein may activate innate immune responses by binding to a pathogen receptor. To confirm that this stimulatory activity is due to the protein and not due to a contaminant, such as endotoxin, DC were treated with RB-nucleocapsid, either intact or digested with V8 protease from *S. aureus*, before adding γδ T cells. Cells were then stained with mAbs specific for cell-surface CD11c, Vδ1 and Vδ2 and intracellular IL-12 and IFN-γ, and the frequencies of CD11c$^+$ cells that produced IL-12 and the frequencies of Vδ1 and Vδ2 T cells that produced IFN-γ were determined by flow cytometry. [Fig 5] shows that RB-nucleocapsid induced significant IL-12 production by DC and IFN-γ production by Vδ1 and Vδ2 T cells and this activity was not abrogated by treatment of the nucleocapsid protein with V8 protease, suggesting that the stimulatory activity in the RB-nucleocapsid protein is due to contaminating non-protein material, such as LPS. To confirm this finding, we also stimulated DC and γδ T cells with a second preparation of nucleocapsid protein (SM-nucleocapsid). SM-nucleocapsid failed to stimulate cytokine production by DC, Vδ1 or Vδ2 T cells ([Fig 5]), confirming that SARS-CoV-2 nucleocapsid protein is not stimulatory for these cells, and that the activity seen with RB-nucleocapsid is due to a non-protein contaminant.

## Immunogenic peptides derived from SARS-CoV-2 spike and nucleocapsid proteins do not stimulate cytokine production by DC, Vδ1 or Vδ2 T cells

DC from 4–5 donors were treated with medium, LPS or overlapping peptides corresponding to the immunodominant epitopes of the spike protein (S peptide), the S1 domain of the spike protein (S1 peptide), the S2 domain (S+ peptide), or the nucleocapsid protein (N peptide). γδ T cells were added after 4 hours and brefeldin A was added 4 hours later. Cells were then stained with mAbs specific for cell-surface CD11c, Vδ1 and Vδ2 and intracellular IL-12 and IFN-γ and the frequencies of CD11c+ cells that produced IL-12 and the frequencies of Vδ1

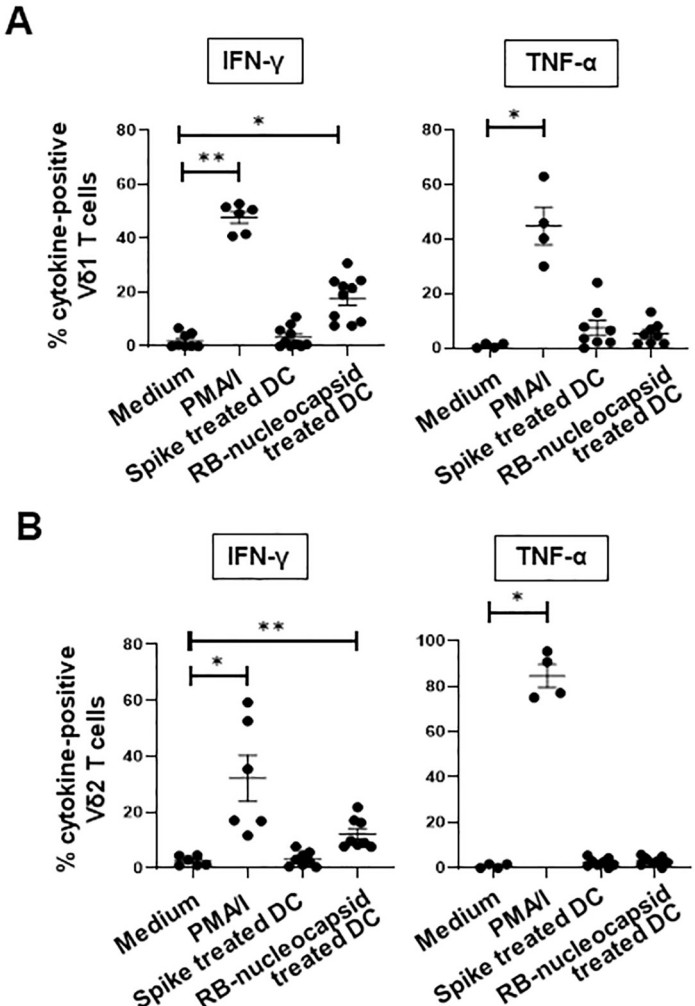

**Fig 4. Effect of dendritic cells treated with SARS-CoV-2 spike and RB-nucleocapsid proteins on cytokine production by Vδ1 and Vδ2 T cells.** γδ T cells from 4–8 donors were co-cultured overnight with allogeneic monocyte-derived DC that had been treated for 4 hours with SARS-CoV-2 spike or RB-nucleocapsid protein. Medium and PMA/I-treated γδ T cells served as negative and positive controls. The percentages of Vδ1 (**A**) and Vδ2 (**B**) T cells that produced IFN-γ and TNF-α in response to each treatment were measured by flow cytometry. Cytokine-positive cell frequencies are compared to those when γδ T cells were treated with medium alone using the Wilcoxon signed-rank test. $^{*}P<0.05$; $^{**}P<0.01$.

and Vδ2 T cells that produced IFN-γ were determined by flow cytometry. Fig 6A shows that none of the peptide pools induced IL-12 production by DC or IFN-γ production by Vδ1 or Vδ2 T cells. However, a mixture of the S and N peptides induced IFN-γ production by 0.33 and 0.26% CD3[+] T cells within PBMC from 2 patients with COVID-19, as shown in Fig 6B, indicating that these peptides are immunogenic for conventional T cells.

## Discussion

γδ T cells are likely to play central roles in immunity against SARS-CoV-2 and/or protection against the immune-mediated pathology associated with severe COVID-19. They frequently exhibit activated/memory phenotypes [7, 19, 20] and accumulate in the lungs of patients with

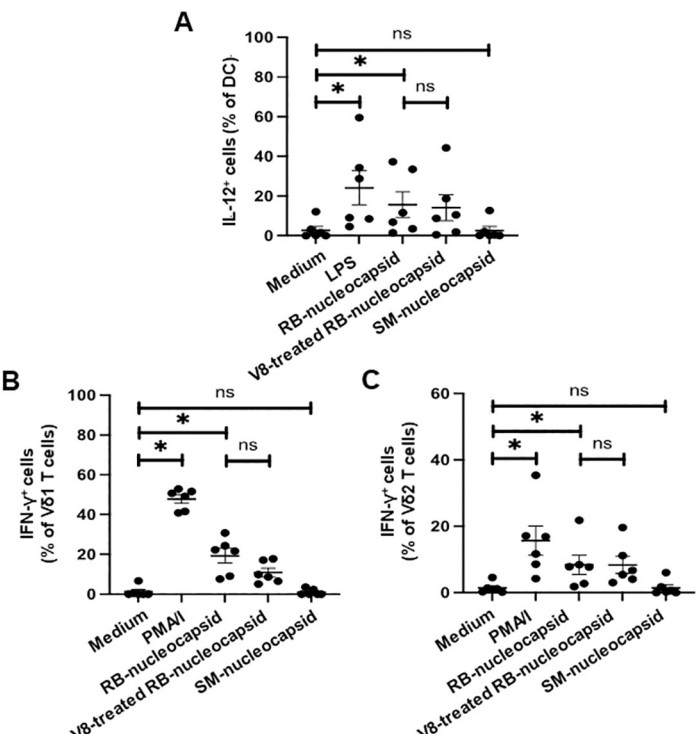

**Fig 5. Activation of DC and γδ by RB-nucleocapsid is mediated by a non-protein contaminant.** Monocyte-derived DC from 6 donors were treated with medium, PMA with ionomycin (PMA/I), RB-nucleocapsid from SARS-CoV-2, V8 protease-treated RB-nucleocapsid and SM-nucleocapsid. γδ T cells were added after 4 hours and the co-cultures were incubated overnight in the presence of brefeldin A. Cells were stained with mAbs specific for cell-surface CD11c, Vδ1 and Vδ2 and intracellular IL-12 and IFN-γ and the frequencies of CD11c+ cells that produced IL-12 (**A**) and the frequencies of Vδ1 (**B**) and Vδ2 (**C**) T cells that produced IFN-γ were determined by flow cytometry. Scatter plots show the percentages of each cell type that produce cytokines in response to each treatment. *P<0.05 using the Wilcoxon signed-rank test. ns, not significant.

severe COVID-19 [7, 22]. Accordingly, they become depleted from the circulation in patients with severe disease [6, 7, 17, 18] and frequently express markers of exhaustion [7, 21], suggesting that their anti-viral activities are suppressed in these individuals. γδ T cells also play central roles in activating and regulating other cells of the innate and adaptive immune system [9, 10, 45–50] and potentially contribute to the inflammatory disease that characterizes severe COVID-19. Therefore, γδ T cells are important potential therapeutic targets for COVID-19.

Central to understanding the roles of γδ T cells in COVID-19 is a knowledge of how these cells become activated. SARS-CoV-2 is a single-stranded RNA virus which produces double-stranded RNA during replication in host cells. This viral RNA is sensed by the innate immune system via cytoplasmic RNA sensors RIG-I and MDA-5 and the endosomal RNA sensors TLR3 and TLR7/8, which lead to the production of proinflammatory cytokines and type 1 interferons [34–37]. Such innate sensing of SARS-CoV-2 RNA occurs in epithelial cells and DC, and both Vδ1 and Vδ2 T cells can express TLR3 and TLR7/8 [29, 51] suggesting the possibility that these cells can also directly recognize SARS-CoV-2. However, it is not known if the virus can infect or be internalized by γδ T cells, which would be a requirement for viral RNA sensing by these cells. Conventional CD4+ T cells can be infected by SARS-CoV-2 [52] and it is possible that the virus can be internalized by γδ T cells via the binding of virus-opsonised antibodies to Fc receptors, such as CD16, which is present on these cells [53].

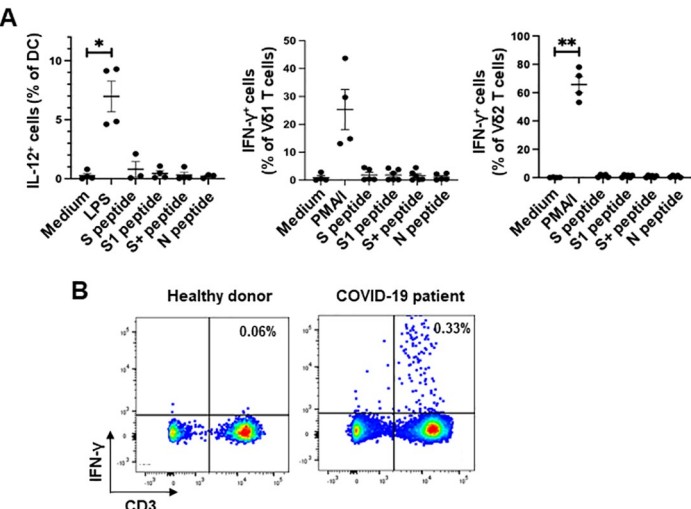

**Fig 6. No activation of dendritic cells or γδ T cells by immunogenic peptides corresponding to SARS-CoV-2 spike and nucleocapsid proteins.** DC from 4–5 donors were treated with medium, LPS or overlapping peptides corresponding to the immunodominant epitopes of the spike protein (S peptide), the S1 domain of the spike protein (S1 peptide), the S2 domain (S+ peptide), or the nucleocapsid protein (N peptide). γδ T cells were added after 4 hours and brefeldin A was added 4 hours later. Cells were then stained with mAbs specific for cell-surface CD11c, Vδ1 and Vδ2 and intracellular IL-12 and IFN-γ and the frequencies of CD11c+ cells that produced IL-12 and the frequencies of Vδ1 and Vδ2 T cells that produced IFN-γ were determined by flow cytometry. **A**, Scatter plots showing percentages of DC that produced IL-12 (left panel) and percentages of Vδ1 (centre) and Vδ2 T cells (right) that produced IFN-γ. *P<0.05; **P<0.01 using the paired t-test. **B**, Flow cytometry dot plots showing IFN-γ production by CD3+ cells within PBMC from a healthy donor and a patient with COVID-19 in response to stimulation for 6 hours with a mixture of the S and N peptides.

γδ T cells can also be activated by ligands produced in response to cellular stress, such as viral infection. Their TCRs recognize non-peptide antigens in a major histocompatibility complex (MHC)-unrestricted manner. The Vδ2 TCR recognises microbial and endogenous pyrophosphate antigens, including isopentenyl pyrophosphate (IPP), which accumulates in tumour cells, making them susceptible to lysis by Vδ2 T cells [10, 54]. If SARS-CoV-2 infection could also lead to IPP production, this would similarly leave infected cells susceptible to killing by these cells. In support of this notion, IPP-stimulated Vδ2 T cells can kill influenza A virus-infected macrophages and inhibit viral replication via IFN-γ production [55, 56]. The Vδ1 TCR recognises a number of stress-inducible molecules expressed by virus-infected and tumour cells, including MICA and MICB [57] and phospholipids and glycolipids presented by CD1 molecules [9, 58] and it is possible that the production of these ligands during SARS-CoV-2 infection could lead to activation of Vδ1 T cells. In addition to activation through the TCR, γδ T cells can also be activated through TLRs, NKG2D and DNAM-1, and by cytokines [10, 13, 29–33]. Therefore, it is conceivable that ligands produced by other immune cells in response to viral infection, such as phosphoantigens, glycolipids, stress-inducible molecules, antibodies or cytokines, could activate γδ T cells.

We investigated if SARS-CoV-2 spike or nucleocapsid proteins can activate cytokine production by γδ T cells, either directly or in the presence of DC. We show that neither spike nor nucleocapsid, nor peptides corresponding to immunodominant regions of these proteins, can directly activate Vδ1 or Vδ2 T cells to produce IFN-γ or TNF-α, either within PBMC or in lines of total γδ T cells. We also report no specific induction of DC maturation or cytokine production by the spike or nucleocapsid proteins or by pools of peptides. Furthermore, no stimulation of IFN-γ or TNF-α production by Vδ1 or Vδ2 in response to treatment of co-cultured

DC with the spike and nucleocapsid proteins or peptides, was observed. Initially, we observed potent IL-12 production by DC and downstream activation of Vδ1 and Vδ2 T cells in response to the SARS-CoV-2 RB-nucleocapsid protein, however, this activity was not abrogated by protease digestion of the nucleocapsid protein and treatment with a second preparation of nucleocapsid protein (SM-nucleocapsid) was found to have no stimulatory activity for DC or γδ T cells. Therefore, the stimulatory activity of the RB-nucleocapsid is likely to be due to contaminating non-proteinaceous material, such as LPS. Interestingly, LPS stimulation of DC led to IL-12 production and downstream activation of Vδ1 and Vδ2 T cells, suggesting that cytokine production by DC contributes to activation of γδ T cells, and that DC activation in response to RNA sensing may similarly activate γδ T cells. In support of this hypothesis, activation of DC with polyinosinic:polycytidylic acid, which is structurally similar to double-stranded RNA, led to IL-12 and TNF-α release by the DC and downstream IFN-γ and TNF-α production by co-cultured γδ T cells (data not shown). Thus, γδ T cell activation in COVID-19 patients is most likely to be secondary to signals produced in response to myeloid cell activation.

Although SARS-CoV-2 spike protein has been demonstrated to bind to TLR4 on macrophages leading to their activation [39, 40], we have found that this protein did not stimulate IL-12 or TNF-α production by monocyte-derived DC. Conversely, LPS, another TLR4 agonist, induced potent cytokine production by DC and downstream activation of γδ T cells. The failure of SARS-CoV-2 spike protein to activate DC in our system, suggests either that macrophages and monocyte-derived DC will respond differently to TLR4 ligation, or that different types of LPS can differentially stimulate DC and macrophages.

In conclusion, we have shown that SARS-CoV-2 spike and nucleocapsid proteins do not activate human Vδ1 and Vδ2 T cells, either directly or via activation of DC. γδ T cell activation in patients with severe COVID-19 is therefore most likely mediated by immune recognition of viral RNA or other structural proteins of SARS-CoV-2, such as the envelope or membrane proteins. Our results show that both Vδ1 and Vδ2 T cells produce IFN-γ in response to co-culture with LPS-stimulated DC and even contaminated RN-nucleocapsid protein, indicating that SARS-CoV-2 activation of DC, and perhaps other cells, may be sufficient to activate γδ T cells in the absence of direct viral recognition by the γδ T cells. Of note, Vδ1 and Vδ2 T cells can reciprocally promote maturation, antigen presentation and cytokine production by DC [10, 44–50], suggesting that γδ T cells and DC may synergise in the immune response to SARS-CoV-2. Future studies using whole SARS-CoV-2 virions, rather than recombinant spike and nucleocapsid proteins are required to determine if the virus can activate γδ T cells directly, or by inducing the production of γδ T cell-stimulatory ligands or cytokines by other innate immune cells, such as DC.

## Supporting information

**S1 File. Raw data for Figs 2–6.**
(XLSX)

## Acknowledgments

The authors are grateful to the donors who participated in this study. We are indebted to the Irish Blood Transfusion Service for providing buffy coat packs as a source of cells for this study.

## Author Contributions

**Conceptualization:** Derek G. Doherty.

**Data curation:** Derek G. Doherty.

**Formal analysis:** Derek G. Doherty.

**Investigation:** Kiran Singh, Sita Cogan, Stefan Elekes, Dearbhla M. Murphy, Sinead Cummins, Rory Curran, Margaret R. Dunne, Gráinne Jameson, Siobhan Gargan, Derek G. Doherty.

**Methodology:** Kiran Singh, Sita Cogan, Stefan Elekes, Dearbhla M. Murphy, Sinead Cummins, Rory Curran, Margaret R. Dunne, Gráinne Jameson, Siobhan Gargan, Derek G. Doherty.

**Resources:** Zaneta Najda, Seamus Martin.

**Supervision:** Aideen Long, Derek G. Doherty.

**Writing – original draft:** Derek G. Doherty.

**Writing – review & editing:** Kiran Singh.

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
