## [Decision Letter · Decision Letter 0]

7 Jun 2022

PONE-D-22-10090SARS-CoV-2 spike and nucleocapsid proteins fail to activate human dendritic cells or γδ T cellsPLOS ONE

Dear Derek,

Thank you for submitting your manuscript to PLOS ONE, and my apologies for the delay in the review process. After careful consideration, we feel that your manuscript has merit but does not fully meet PLOS ONE’s publication criteria as it currently stands. Therefore, we invite you to submit a revised version of the manuscript that addresses the points raised by the reviewers, as summarised at the bottom of this letter.

In particular, a revised manuscript should place a stronger emphasis on the physiological relevance of the experiments performed and the underlying hypothesis of a direct action of human γδ T cells by viral proteins, considering alternative mechanisms by which human γδ T cells might be activated in the context of a SARS-CoV-2 infection.

Please submit your revised manuscript and your reply to the reviewers' comments by Jul 01 2022 11:59PM. If you need more time than this to complete your revisions, please reply to this message or contact the journal office at plosone@plos.org. Please include the following items when submitting your revised manuscript:

We look forward to receiving your revised manuscript.

Kind regards,

Matthias Eberl, Ph.D.

Academic Editor

PLOS ONE

Journal Requirements:

Reviewers' comments:

Reviewer's Responses to Questions

**Comments to the Author**

1. Is the manuscript technically sound, and do the data support the conclusions?

Reviewer #1: Yes

Reviewer #2: Yes

Reviewer #3: Partly

2. Has the statistical analysis been performed appropriately and rigorously? 

Reviewer #1: Yes

Reviewer #2: Yes

Reviewer #3: Yes

3. Have the authors made all data underlying the findings in their manuscript fully available?

Reviewer #1: Yes

Reviewer #2: Yes

Reviewer #3: Yes

4. Is the manuscript presented in an intelligible fashion and written in standard English?

Reviewer #1: Yes

Reviewer #2: Yes

Reviewer #3: Yes

5. Review Comments to the Author

Reviewer #1: This manuscript addresses a clear knowledge gap with a logical sequence of robust experiments from which appropriate conclusions have been drawn. There is considerable interest in the role of γδ T cells in mucosal inflammatory disorders, particularly in the context of COVID-19 and the development of new drugs to combat this. It is therefore important to understand the mechanisms that drive γδ T cells activation and recruitment to the lung. In this report, the authors present strong evidence that γδ T cells are not directly activated by the major viral spike or capsid proteins, which has obvious implications for efforts to target this population in the clinic. The findings are technically sound, described clearly, and represent an important contribution to the growing body of knowledge in this area.

Reviewer #2: This manuscript investigates whether proteins or peptides from SARS-Cov2 spike and nucleocapsid activate Vd1 or Vd2 T cells. While it is known that gd T cells in humans respond and participate in the immune response to SARS-Cov2, it is still unknown how they become activated. While the data here represent a start to this endeavor, there are very few conclusive findings in the manuscript in it's current form.

1. There is little investigation or discussion into the very possible point that the Vd2 T cells respond to upregulation of IPP upon virus infection of neighboring cells. Direct viral protein activation in human gd T cells is not a common mechanism of activation, so seems a strange hypothesis. Activation of gamma delta T cells by isoprenyl phosphate molecules has been carefully examined with regard to influenza virus and may be a stronger hypothesis here, especially with the negative findings.

2. There was contamination of the proteins used in the study negating the results from several figures. These results should not be included.

3. There was no positive control to support spike or nucleocapsid proteins being present or at the proper concentrations etc.

4. The findings are minimal and more studies need to be performed to come to any conclusions.

Reviewer #3: This manuscript concludes that Spike protein and NP from SARS-Cov2 do not result in activation of d1 or de gd T cells, whether directly or through DC. Minor DC-mediated activation is hypothesised to be through contaminant, such as viral RNA. The paper is technically ok, well written and the data well described. Whilst the conclusions from the studies performed are correct there are questions as to the relevance of these in a physiological in vivo context: gdT cells (d1 as well as d2) sense infected cells (viral as well as bacterial) and whilst the authors point this out in the discussion, they did not address this by studying gdT reactivity following exposure to infected cells, DC or other. The other possible means by which gdT cells can be activated is if exposed to antibody opsonised virus or infected cells. Such Ab and target can then bind to FcR (CD16) on the gdT cells and lead to gd T activation and target uptake. This was not studied or even mentioned as possibility. These two points diminish the value of the work.

No minor points - very good figures, legends and text.

6. PLOS authors have the option to publish the peer review history of their article (what does this mean?). If published, this will include your full peer review and any attached files.

Reviewer #1: No

Reviewer #2: No

Reviewer #3: No

---

## [Author Response · Author response to Decision Letter 0]

17 Jun 2022

Full responses to all comments from the editors and reviewers can be found in the uploaded document entitled 'Singh et al. PLoS One - responses to reviewers'

---

## [Decision Letter · Decision Letter 1]

1 Jul 2022

SARS-CoV-2 spike and nucleocapsid proteins fail to activate human dendritic cells or γδ T cells

PONE-D-22-10090R1

Dear Dr. Doherty,

We’re pleased to inform you that your manuscript has been judged scientifically suitable for publication and will be formally accepted for publication once it meets all outstanding technical requirements.

Kind regards,

Matthias Eberl, Ph.D.

Academic Editor

PLOS ONE

Additional Editor Comments (optional):

Reviewers' comments:

Reviewer's Responses to Questions

**Comments to the Author**

1. If the authors have adequately addressed your comments raised in a previous round of review and you feel that this manuscript is now acceptable for publication, you may indicate that here to bypass the “Comments to the Author” section, enter your conflict of interest statement in the “Confidential to Editor” section, and submit your "Accept" recommendation.

Reviewer #1: All comments have been addressed

Reviewer #2: All comments have been addressed

2. Is the manuscript technically sound, and do the data support the conclusions?

Reviewer #1: Yes

Reviewer #2: Yes

3. Has the statistical analysis been performed appropriately and rigorously? 

Reviewer #1: Yes

Reviewer #2: Yes

4. Have the authors made all data underlying the findings in their manuscript fully available?

Reviewer #1: Yes

Reviewer #2: Yes

5. Is the manuscript presented in an intelligible fashion and written in standard English?

Reviewer #1: Yes

Reviewer #2: Yes

6. Review Comments to the Author

Reviewer #1: All comments have been fully addressed / no further concerns. The revised manuscript is acceptable for publication.

Reviewer #2: (No Response)

7. PLOS authors have the option to publish the peer review history of their article (what does this mean?). If published, this will include your full peer review and any attached files.

Reviewer #1: No

Reviewer #2: No

---

## [Editor Report · Acceptance letter]

5 Jul 2022

PONE-D-22-10090R1 

SARS-CoV-2 spike and nucleocapsid proteins fail to activate human dendritic cells or γδ T cells 

Dear Dr. Doherty:

I'm pleased to inform you that your manuscript has been deemed suitable for publication in PLOS ONE. Congratulations! Your manuscript is now with our production department. 

Kind regards, 

on behalf of

Professor Matthias Eberl 

Academic Editor

PLOS ONE